# The binding mechanism of an anti-multiple myeloma antibody to the human GPRC5D homodimer

Pengfei Yan [1,2,5], Xi Lin [1,5], Lijie Wu [1], Lu Xu [1,3], Fei Li [1], Junlin Liu[1] & Fei Xu [1,2,4] ✉

GPRC5D is an atypical Class C orphan G protein-coupled receptor. Its high expression on the surface of multiple myeloma cells has rendered it an attractive target for therapeutic interventions, including monoclonal antibodies, CAR-T cells, and T-cell engagers. Despite its therapeutic potential, the insufficient understanding regarding of the receptor's structure and antibody recognition mechanism has impeded the progress of effective therapeutic development. Here, we present the structure of GPRC5D in complex with a preclinical-stage single-chain antibody (scFv). Our structural analysis reveals that the GPRC5D presents a close resemblance to the typical Class C GPCRs in the transmembrane region. We identify a distinct head-to-head homodimer arrangement and interface mainly involving TM4, setting it apart from other Class C homo- or hetero-dimers. Furthermore, we elucidate the binding site engaging a sizable extracellular domain on GPRC5D for scFv recognition. These insights not only unveil the distinctive dimer organization of this unconventional Class C GPCR but also hold the potential to advance drug development targeting GPRC5D for the treatment of multiple myeloma.

Multiple Myeloma (MM) ranks as the second most common hematologic malignancy among prevalent malignant blood system tumors. Its primary manifestation involves the uncontrolled proliferation and accumulation of monoclonal plasma cells within the bone marrow, resulting in excessive production of immunoglobulins, bone resorption, and damage to end-organ systems[1–4]. Despite the recent development of numerous antibody-based therapies, including bispecific antibodies[5], CAR-T cell therapy[6], and antibody-drug conjugates (ADC)[7], only a small portion of these antibody drugs have entered clinical use[8,9]. MM remains largely an incurable malignancy at present[5,10–12].

The GPRC5D is an orphan receptor, without an identified endogenous ligand. GPRC5D belongs to the Class C G protein-coupled receptors (GPCRs) within the group 5 subfamily, member D from which it derives its name. In mammals, there are three other GPRC5 subfamily members: GPRC5A, GPRC5B, and GPRC5C[13–16]. While the conventional Class C GPCRs such as mGluR usually contains a large N-terminal domain named Venus flytrap (VFT) module responsible for orthosteric ligand binding and obligate homo- or hetero-dimer formation[17–19]; GPRC5D, as well as the entire GPRC5 subfamily, possesses relatively small extracellular domains, comprising only 20–50 amino acids (Fig. 1a and Supplementary Fig. 1a). It was suggested by some early research that despite GPRC5D's homology with the Class C receptors, its topology might resemble more closely to that of Class A receptors[13]. Furthermore, due to the absence of a substantial N-terminal extracellular region, questions persist regarding whether GPRC5D can still form dimers and how the signal is transduced through the transmembrane region.

In the human body, GPRC5D is predominantly expressed in the nails and hair, or more precisely, it is exclusively expressed in

[1]iHuman Institute, ShanghaiTech University, Shanghai, China. [2]School of Life Science and Technology, Shanghai Key Laboratory of High-resolution Electron Microscopy, ShanghaiTech University, Shanghai, China. [3]JiKang Therapeutics, Shanghai, China. [4]Shanghai Clinical Research and Trial Center, Shanghai, China. [5]These authors contributed equally: Pengfei Yan, Xi Lin. ✉e-mail: xufei@shanghaitech.edu.cn

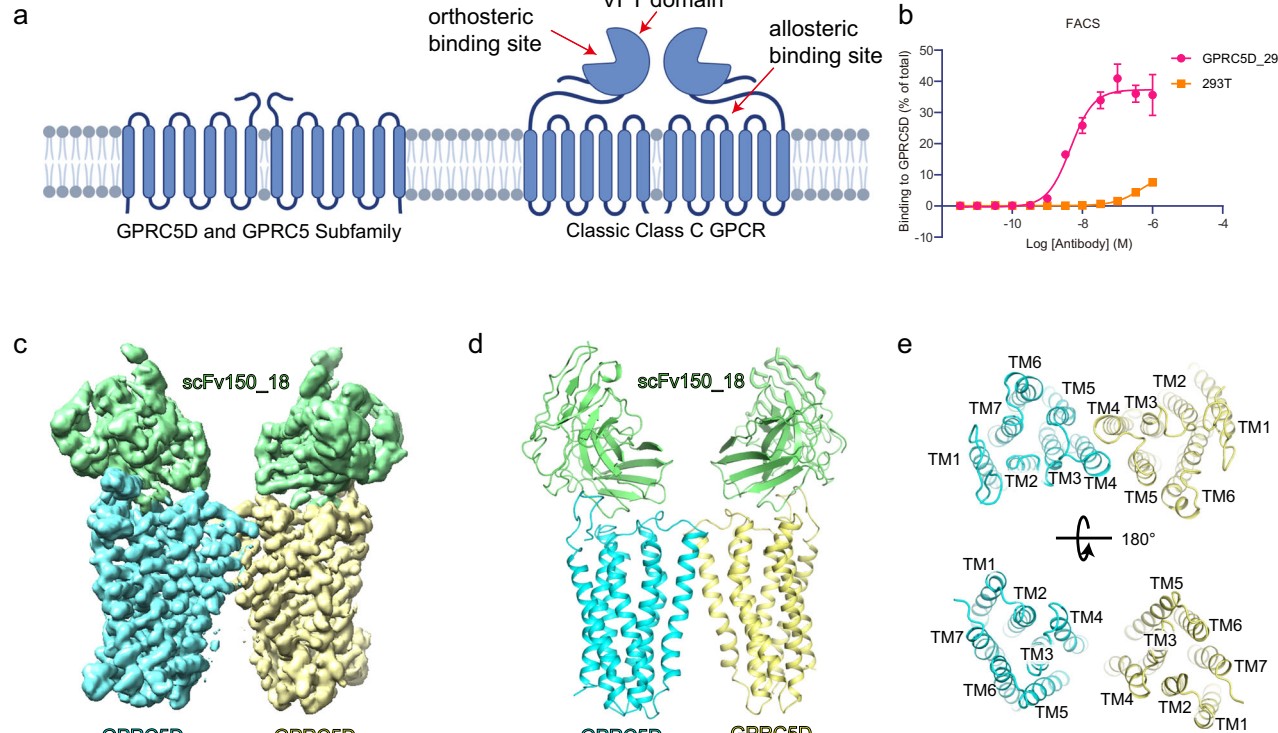

**Fig. 1 | Cryo-EM structure of the GPRC5D and scFv150_18 complex. a** The schematic model of GPRC5D, characterized by a short extracellular N-terminus and lack of VFT domain, in comparison to classical Class C GPCRs. Figure 1a Created with BioRender.com released under a Creative Commons Attribution-NonCommercial-NoDerivs 4.0 International license https://creativecommons.org/licenses/by-nc-nd/4.0/deed.en. **b** The binding capacity of the antibody (scFv150_18 in its IgG format) was measured using Fluorescence-activated cell sorting (FACS) assay (see "Methods") on GPRC5D-expressing cell surface. 293 T denotes the control without GPRC5D transfection. Data were represented as the mean ± s.e.m. and from three independent experiments. **c, d** Cryo-EM map (**c**) and model (**d**) of the GPRC5D and scFv150_18 complex. Yellow and cyan indicate the two GPRC5D subunits forming a dimer, while green represents the scFv bound to it. **e** Extracellular and intracellular views of GPRC5D structure, transmembrane helices TM1–TM7 are labeled.

differentiating cells that produce hard keratin proteins[15]. Recent research has revealed the highly specific and augmented expression of GPRC5D on the surface of multiple myeloma cells, thereby establishing it as an attractive target for antibody-based therapies against MM[20]. As of now, 17 antibody drugs targeting GPRC5D have entered clinical trials, capitalizing on its tumor marker feature on the cell membrane. On August 10, 2023, Johnson & Johnson announced the FDA approval of TALVEY™ (talquetamab-tgvs)[5,9,21], the world's first bispecific antibody targeting GPRC5D/CD3, for the treatment of relapsed or refractory multiple myeloma in adult patients[5]. The potential for targeting GPRC5D continues to show extensive promise.

Here, we elucidate the structure of the complex formed between human GPRC5D and a single-chain antibody (scFv) using cryo-electron microscopy. Through screening of a series of GPRC5D antibodies from the published patents in different formats (Fab, scFv, etc.), we identified a stable complex formed by one of the single-chain fragments derived from the patented scFv150-18 (US20200123249A1)[22]. The molecular interface between the antibody and GPRC5D is well-defined. The structural analysis revealed that similar to the conventional Class C GPCRs, GPRC5D also forms a homodimer but exhibits a unique dimerization organization. Combined with mutagenesis and functional experiments, we aim to leverage these structural insights to guide the development of antibody therapeutics targeting GPRC5D, addressing the current gap in the treatment of multiple myeloma.

## Results
### The overall structure of the scFv150-18-GPRC5D complex
To guide the design and development of new-generation antibody therapeutics for treating multiple myeloma, we attempted to elucidate the structure of an antibody in complex with the human GPRC5D. The antibody (scFv150-18) used in this study was derived from a preclinical-stage drug candidate as illustrated in the patent[22] (US20200123249A1). We selected two antibody sequences from the patent and cloned them into scFv and Fab formats. Subsequently, we screened these four antibodies in an attempt to obtain stable complexes with GPRC5D. We explored various methods including the co-expression of antibody and GPRC5D in insect cells as well as incubation of the purified antibody and GPRC5D proteins (Supplementary Fig. 2). Finally, we successfully obtained a well-formed complex between GPRC5D and the scFv (scFv150-18) through the co-expression approach. To profile the antibody binding to GPRC5D, we cloned the scFv into human IgG format (IgG150-18) and conducted fluorescence-activated cell sorting (FACS) assays. The result showed that IgG150-18 could bind to GPRC5D extracellular domain with an EC50 of approximately 4.6 nM (Fig. 1b). To overcome the low surface expression issue of the wild-type hGPRC5D protein, we employed a fusion approach by attaching the thermostabilized apocytochrome b562RIL (BRIL) to the receptor's N terminus[23] ("Methods" and Supplementary Fig. 1a, b). Co-expression of the antibody and GPRC5D in the *Trichuplusia ni Hi5* insect cells yielded a stable complex amenable to cryo-EM study. Size-exclusion chromatography (SEC) and SDS-PAGE analysis revealed that GPRC5D can form a monodispersed complex with scFv150-18 (Supplementary Fig. 3a). We finally determined the cryo-EM structure of GPRC5D-scFv150-18 complex with nominal global map at 3.34 Å resolution (Fig. 1c–e, Supplementary Fig. 3b–f and Supplementary Table 1). The overall structure of GPRC5D adopts a canonical seven-transmembrane (7TM) fold like classical GPCRs.

## The extensive binding interface between scFv and GPRC5D

The antibody whose structure was resolved in this study is derived from a monoclonal antibody patent targeting GPRC5D (US20200123249A1)[22]. Our structure unveils that the antibody extensively binds to the extracellular regions in GPRC5D, including the N-terminal region and all three extracellular loops (ECL1, ECL2, and ECL3), with an overall interface area of 911 Å$^2$ within one copy of the GPRC5D-scFv complex or 1822 Å$^2$ for the GPRC5D homodimer (Fig. 2a). The binding between the extracellular region of GPRC5D and the antibody is primarily mediated by three pairs of hydrogen bonds: Y226$^{HCDR2}$ with S8$^{N-term}$, Y228$^{HCDR2}$ with the main-chain carbonyl oxygen of C16 $^{N-term}$ and T227$^{HCDR2}$ with the main-chain carbonyl oxygen of Q236$^{ECL3}$. Other polar and hydrophobic interactions contribute to the interface as well (Fig. 2b).

To further validate the antibody binding interface, we introduced combined mutations in the N-terminal region (S8G, D11G, L15G, C16G, D17G), ECL1 (E83G, L84G), ECL2 (R154G, V159G, N160G, T162G, P163G, L166G), ECL3 (P235G, Q236G), as well as the entire interface mutations on GPRC5D. To examine the antibody binding to GPRC5D in a cellular system, we purified the antibody in the IgG format (IgG150-18) and conducted FACS assays on cells overexpressing WT GPRC5D or the mutants. The results indicated a significant reduction in binding capacity for various mutants in comparison to the WT GPRC5D. Notably, mutations in the N-terminus almost abolished antibody binding, consistent with our structural observation that the scFv primarily binds to the N-terminus of GPRC5D (Fig. 2c). To further delineate the specific recognition between GPRC5D and the antibody, we engineered single mutations on proximal residues within both the receptor and the antibody. The FACS assay results demonstrate that mutations D11A, L15A, and C16A at the N-terminus of GPRC5D markedly impaired the antibody binding (Fig. 2d). Notably, antibody binding was significantly diminished by a single mutation on Y228$^{HCDR2}$, which interacts with L15 and C16 at the N-terminus of GPRC5D (Fig. 2d). In addition, mutations of residues involved in hydrogen bonding, such as Q236$^{ECL3}$ of GPRC5D and T227$^{HCDR2}$ of the antibody, substantially impaired antibody binding, underscoring the importance of these hydrogen bonding interactions (Fig. 2d). Collectively, our structural elucidation of the molecular interactions between the antibody and GPRC5D, complemented by mutagenesis and functional data, validates the antibody binding mode. This structural insight into the GPRC5D-antibody interaction can guide the design and development of better antibody therapeutics.

## An atypical Class C GPCR

Previous studies predicted that despite GPRC5D's homology with the Class C GPCRs, its topology resembles closer to Class A GPCRs owing to its relatively short N-terminal extracellular region[13]. To verify this hypothesis, we conducted extensive sequence alignments and structural comparisons to determine whether GPRC5D exhibits greater similarity to the Class A or the Class C GPCRs. Our findings reveal that the full-length GPRC5D shows low sequence similarity with the Class A receptors (less than 22%). Even when comparing only the transmembrane helices (excluding the N and C-terminal regions and interhelical loops), the sequence similarity remains below 29%. In contrast, when comparing GPRC5D with the Class C GPCRs, considering GPRC5D's limited N-terminal extracellular region, we focused solely on the transmembrane region. We observed relatively high sequence similarities within the GPRC5 subfamily (62% − 73%). Moreover, the sequence similarity of GPRC5D with other Class C receptors ranges from 29% to 47%, which is notably higher than that with Class A receptors (Supplementary Fig. 1c, d).

More importantly, our structural analysis reveals that despite lacking a large N-terminal extracellular VFT domain, the transmembrane domain of GPRC5D resembles closer to the Class C receptors. When compared to Class A GPCRs, we observe large-scale helical shift on TM3 and TM5 of GPRC5D which move to the center of the 7TM bundle (Fig. 3a and Supplementary Fig. 4a). This is consistent with the previous observation in the 7TM structure of mGluR$_1$[24]. A similar trend on helical shift was observed when compared to Class B and Class F GPCRs (Fig. 3b, d and Supplementary Fig. 4b, d). However, upon comparison with other Class C GPCRs, it is evident that the transmembrane helical region of GPRC5D exhibits overall similarity, except for TM3, which is positioned closer to the helical center (Fig. 3c and Supplementary Fig. 4c). Based on the structural observations, we compared the RMSD values in the transmembrane regions between GPRC5D and other GPCRs. It is notable that, in comparison with the Class C receptors, the overall RMSD values fall within the range of 1 Å to 8 Å, although GPR158 exhibits larger variations with RMSD values exceeding 10 Å. However, when compared with the Class A receptors, the RMSD values in the transmembrane region are generally higher than 10 Å (Supplementary Table 2). This unique helical arrangement in GPRC5D may suggest a distinctive G protein binding pattern, although it remains vague whether GPRC5D signals through the canonical G protein pathways and which G protein subtype it may couple to ref.[25].

In summary, our findings suggest that despite GPRC5D having a short N-terminal extracellular region resembling Class A GPCRs, both its sequence and structural features show a more significant resemblance to the Class C GPCRs. In addition, the relatively lower sequence similarity and structural alignment emphasize the distinct nature of GPRC5D within the Class C GPCRs.

## A shallow pocket in the transmembrane region and inactive-like conformation

In the absence of an agonist or antagonist in the orthosteric pocket and the lack of downstream G proteins, the conformational state of the GPRC5D structure remains undetermined. Therefore, we conducted an extensive comparison of GPRC5D with other GPCR structures (Fig. 3c and Supplementary Fig. 4c). We first examined the transmembrane pocket which was present in conventional GPCRs including orphan receptors. GPRC5D is categorized as an orphan GPCR, and as of now, no small molecule ligands have been reported. Consequently, we did not detect any additional cryo-EM density in the transmembrane pocket that could be attributed to any unknown ligands. Thus, we have confirmed that the structure is in a ligand-free state. This is in contrast to the scenarios in some other orphan receptors. For example, in the orphan GPR20[26] and GPR88[27], unassigned density was observed in the orthostatic pocket; in the orphan GPR119[28] and GPR84[29], the lipid density was observed. Instead, we observed an "unfavorable" ligand binding pocket in GPRC5D (Supplementary Fig. 5a−e). Structural alignment of GPRC5D's pocket with other Class C GPCRs revealed the potential clashes between F$^{5.40}$-W$^{6.53}$ pair of GPRC5D with virtually all small molecule ligands within the transmembrane pockets of all reported Class C receptors (Supplementary Fig. 5f−i). Examination of these two residues within the pocket of GPRC5D, it became evident that they may obstruct the ligand, if any, from extending to the helical core (Supplementary Fig. 5e). Sequence alignment reveals that residues 5.40 and 6.53 are highly hydrophobic across the Class C family, with residue 6.53 almost universally aromatic. Moreover, the conservation of these two residues within other GPRC5 receptors suggests their potential role in shaping a similarly shallow pocket for this subfamily (Supplementary Fig. 5j).

To investigate the functional state of the antibody-bound and ligand-free GPRC5D structure, we next conducted structural alignments to inactive and active structures of all reported Class C GPCRs. To optimize alignment quality, we focused on the transmembrane region and calculated the RMSD values. The results reveal that the structural similarity between GPRC5D and other orphan GPCRs in the Class C family is notably low, such as with the apo state of GPR158[30] (PDB: 7EWL; RMSD: 16.389 Å). Instead, the closest resemblance is observed with the inactive state of the mGluR$_2$[31] (Supplementary Table 2). We conducted a

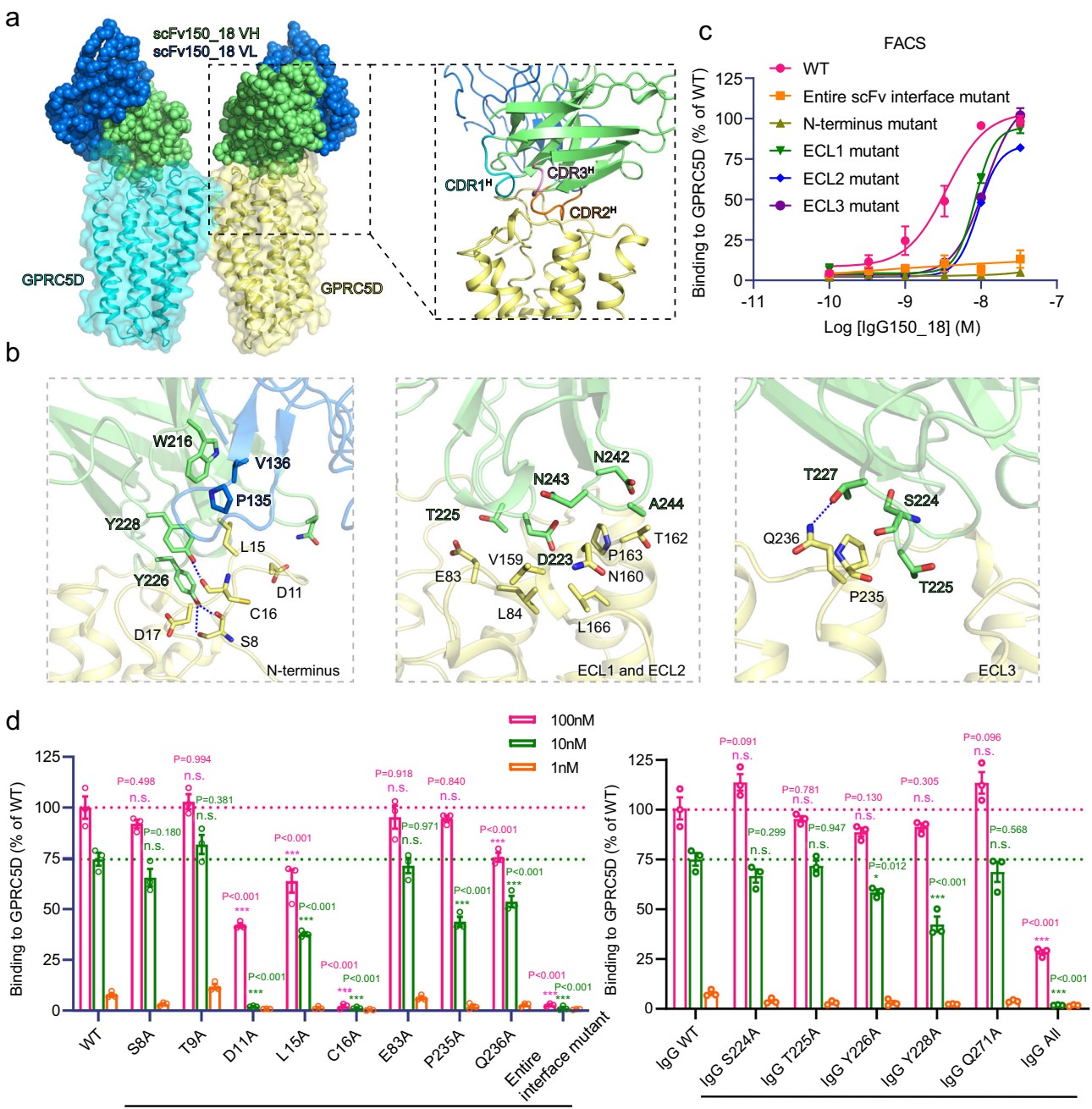

**Fig. 2 | The interface between GPRC5D and scFv150_18. a** The overall structure of the GPRC5D and scFv150_18 complex. GPRC5D is shown in a cartoon overlaid with a transparent surface; the two subunits of the GPRC5D dimer are shown in cyan (left) and yellow (right), respectively. The scFv is shown in a sphere representation, with the light chain depicted in blue and the heavy chain in green. **b** The molecular interactions between scFv and GPRC5D within three regions: the N-terminal extracellular region (left), ECL1 and ECL2 (middle), and ECL3 (right), respectively. **c** FACS assay to measure the binding capacity of the antibody to various scFv interface mutants. The term 'entire scFv interface mutant' refers to the combined mutation of all identified interface residues on GPRC5D to glycine. N-terminus mutant (S8G, D11G, L15G, C16G, D17G), ECL1 mutant (E83G, L84G), ECL2 mutant (R154G, V159G, N160G, T162G, P163G, L166G), and ECL3 mutant (P235G, Q236G) denote the combined mutations of interacting residues on N-term, ECL1, ECL2, and ECL3 to glycine, respectively. Data were represented as the mean ± s.e.m. ($n = 3$) and normalized to WT GPRC5D. **d** FACS binding assays were performed using mutant antibodies or mutant GPRC5D. The term "entire scFv interface mutant" denotes combined mutations at all scFv binding sites on GPRC5D, whereas "IgG all" denotes combined mutations including S244A, T225A, Y226A, Y228A, and Q271A on IgG. Data were represented as the mean ± s.e.m. ($n = 3$) and normalized to WT GPRC5D. Significance was determined by Ordinary one-way analysis of variance (ANOVA), followed by Dunnett's multiple comparisons test (***$P < 0.001$, *$P < 0.05$; n.s., not significant).

structural comparison of GPRC5D with both the active[32] (PDB: 7E9G) and inactive[31] (PDB: 7EPE) states of mGluR$_2$. GPRC5D aligns closely with the TM6 of the inactive-state structure of mGluR$_2$. In mGluR$_2$, the TM6 undergoes a downward shift by half a helix turn from inactive to active states (Fig. 3e and Supplementary Fig. 6b). Previous studies suggest that

this TM6 movement is a characteristic of Class C receptor activation[31]. Similarly, this downward movement of TM6 is also observed in the active-state mGluR$_5$ (Supplementary Fig. 6d, e). A closer comparison of GPRC5D with mGluR$_2$ revealed a noteworthy rotational conformational change in the highly conserved residue W$^{6.50}$ of the Class C GPCR. In

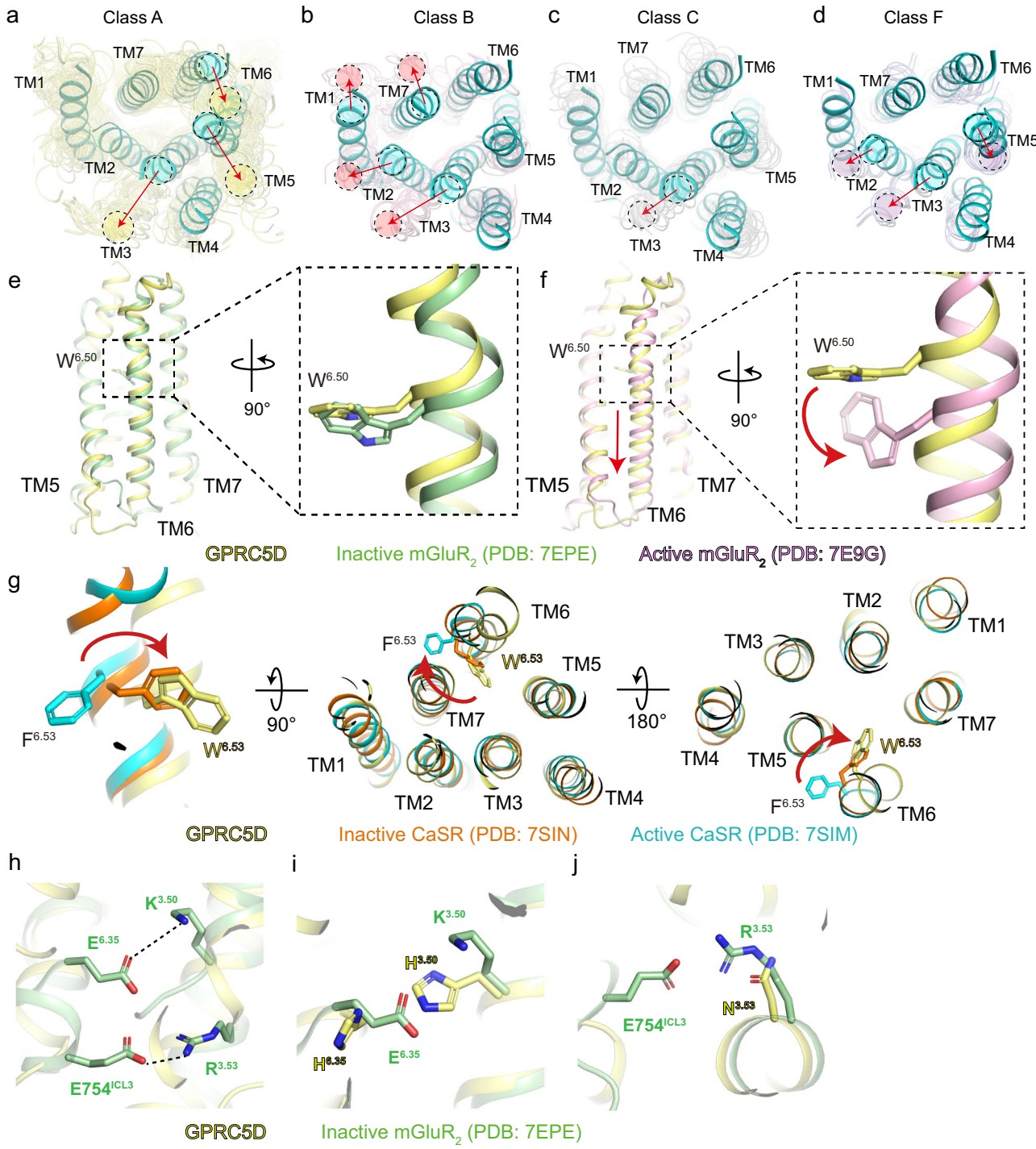

**Fig. 3 | The comparative analysis of the 7TM structure of GPRC5D with other representative class of GPCRs.** Extracellular view of superimpositions between GPRC5D shown in cyan and (**a**) class A GPCRs shown in yellow, (**b**) class B GPCRs shown in pink, (**c**) class C GPCRs shown in gray, and (**d**) class F GPCRs, shown in purple. In (**a–d**), shifts of each TM in GPRC5D relative to other classes are indicated by red arrows. **e, f** Comparison between the TM6 of GPRC5D and the TM6 of inactive and active states of mGluR$_2$. GPRC5D is in yellow, the inactive state of mGluR$_2$ is in green, and the active state of mGluR$_2$ is in pink. **g** The comparison of residue 6.53 between inactive CaSR (orange, PDB: 7SIN), active CaSR (cyan, PDB: 7SIM), and GPRC5D (yellow). **h–j** The two ionic locks, E$^{6.35}$-K$^{3.50}$, and E$^{ICL3}$-R$^{3.53}$, in the inactive mGluR$_2$ (green), and the corresponding residues in GPRC5D (yellow) are indicated.

GPRC5D, its W$^{6.50}$ residue resembles that in the inactive state of mGluR$_2$ (Fig. 3e, f and Supplementary Fig. 6c). We next examined residue W$^{6.53}$ in GPRC5D, which points towards the core of the TM bundle, similar to the position of the key activation-related residue F$^{6.53}$ in CaSR[33] in its inactive state (Fig. 3g). However, it is noteworthy that Class C receptors exhibit variations in the movements of TM6 during activation. For instance, TM6 exhibits upward movement in mGluR$_2$[34] and CaSR[35], and outward

rotation in mGluR$_5$[36]. Although we have comparatively analyzed the structural features of GPRC5D, its activation mechanism remains to be further investigated due to the lack of an active state structure and the fact that it is an atypical Class C GPCR. Taken together, we propose that GPRC5D adopts an inactive-like conformation, despite the overall low sequence similarity of activation-related motifs between GPRC5D and other Class C receptors.

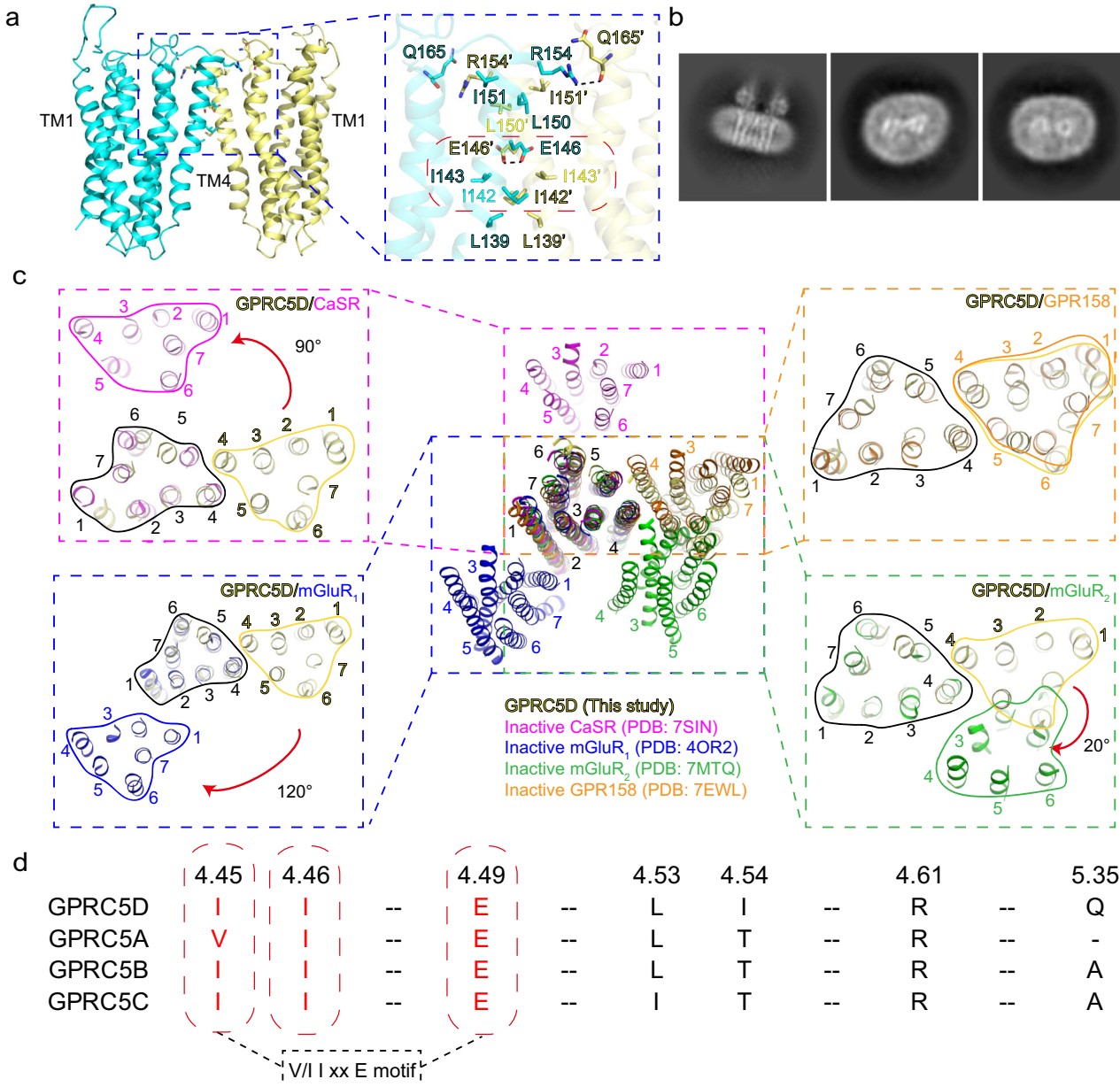

**Fig. 4 | A unique dimer interface. a** The dimer interface in GPRC5D. The two subunits are colored in yellow and cyan. On the right is a zoomed-in view of the interface with key residues labeled. The molecular interactions within the "V/I I x x E" motif are highlighted with the red box. **b** Representative images for the cryo-EM 2D classification averages. The images, from left to right, depict the side view, extracellular view, and intracellular view, respectively. **c** The comparison of dimer interfaces in GPRC5D with other inactive-state Class C GPCR structures (CaSR, mGluR$_1$, mGluR$_2$, and orphan GPR158), viewed from the extracellular side. Seven helices are labeled. One subunit of the dimer is superimposed with the boundary labeled in black, the movement on the other subunit (boundary labeled in yellow for GPRC5D and the annotated color for the comparing receptor) relative to GPRC5D is indicated by the red arrow. **d** Dimer interface sequence alignment in the GPRC5 subfamily. The red box indicates the consensus dimer interface motif in the GPRC5 subfamily.

Class C GPCRs exhibit a highly conserved salt bridge between residues $K^{3.50}$ and $E^{6.35}$, commonly referred to as the "ionic lock" in Class C. In addition, there is another ionic interaction formed by the less-conserved residues $R^{3.53}$ and $E^{ICL3}$ in some Class C GPCRs such as mGluR$_2$[24,31,37]. Some studies suggest that these ionic interactions stabilize the inactive conformation of mGluR$_2$[31] (Fig. 3h). We conducted sequence and structural alignment and observed that residues 3.50 are H/Q and 6.35 are H/N within the GPRC5 subfamily which can no longer form ionic interactions (Supplementary Fig. 6g). Meanwhile, the ionic pair $R^{3.53}$-$E^{ICL3}$ in mGluR$_2$ lacks conservation throughout the GPRC5 subfamily. Furthermore, structural analysis indicates that in GPRC5D, residues 3.50 and 6.35 are distantly positioned, making their interaction impossible (Fig. 3i, j). Overall, these findings suggest that

while GPRC5D appears to be in an inactive state, its conformation differs notably from other Class C receptors, highlighting its uniqueness.

## A distinctive dimer interface of GPRC5D

Despite lacking a large extracellular VFT domain and exhibiting low sequence similarity with other Class C GPCRs, GPRC5D forms a homodimer, reminiscent of classical Class C receptors (Fig. 4a). Notably, no monomeric species were observed in the cryo-EM 2D classification averages (Fig. 4b and Supplementary Fig. 3c). The dimer interface is primarily located within TM4, adopting a head-to-head configuration similar to the recently reported APJR homodimer[38]. The area of the dimer interface is 683.76 Å$^2$, much larger than that in the

APJR dimer (140 Å$^2$)[38]. While the dimer interfaces of most Class C receptors are located in the N-terminal extracellular VFT domain, GPRC5D uniquely relies on the transmembrane region for its dimer interface. Structural and functional analysis of the dimer interface revealed that it is predominantly mediated by two pairs of hydrogen bonds and a series of hydrophobic interactions. The hydrogen bonds involve E146$^{4.49}$ from the two symmetric GPRC5D molecules, as well as R154$^{4.61}$ with Q165$^{5.35}$ in each molecule. Meanwhile, the hydrophobic interactions encompass the hydrophobic amino acids I142$^{4.45}$, I143$^{4.46}$, L150$^{4.53}$, and I151$^{4.54}$ (Fig. 4a).

Class C GPCRs typically function through the formation of homodimers or heterodimers. In addition, given that our structure is in an inactive-like state, we concentrated on comparing the dimer interface area (Supplementary Table 3) and dimer arrangement (through alignment on one subunit of the dimer to compare the orientation of the other subunit, see Supplementary Fig. 7a) of representative Class C GPCRs in their inactive states. When comparing GPRC5D with the inactive state of mGluR$_1$[24], sharp differences on the dimer arrangement were observed. While GPRC5D adopts a head-to-head dimer configuration, the mGluR$_1$ displays a tail-to-tail arrangement at the transmembrane region. In addition, mGluR$_1$ shows an approximate 120° rotation in the entire dimer interface compared to GPRC5D, primarily due to its dimer interface residing within TM1. Similarly, the inactive CaSR[33] adopts a tail-to-tail configuration in its dimer interface, with primary interactions occurring within TM5 (Fig. 4c).

It's noteworthy that the dimer interface of Class C orphan GPR158[30,39] closely resembles that of GPRC5D, adopting a head-to-head configuration with primary interactions occurring within TM4 (Fig. 4c). Further comparison of amino acids at the GPRC5D and GPR158 dimer interfaces reveals sharp differences. The above-mentioned hydrogen bonding and hydrophobic interactions important for stabilizing GPRC5D dimer are not conserved in GPR158 (Supplementary Fig. 7c).

To further examine the uniqueness of the GPRC5D dimer interface, we conducted consensus analysis across Class C GPCRs. It's evident that these residues constituting the dimer interface in GPRC5D lack conservation within the Class C GPCRs while displaying certain level of conservation within the GPRC5 subfamily (Supplementary Fig. 7c). Based on this result, we propose a motif V/I$^{4.45}$I$^{4.46}$xxE$^{4.49}$, which might represent a conserved dimer interface motif within the GPRC5 subfamily (Fig. 4d).

## Discussion

Research on GPRC5D is currently in its early stages, with downstream signaling pathways and endogenous ligands remaining unknown. However, its high expression on the surface of multiple myeloma cells has positioned it as a prominent therapeutic target for multiple myeloma treatment[20]. Numerous monoclonal antibodies, including bispecific antibodies, trispecific antibodies, and antibody-drug conjugates are under development[5–7,10–12,40]. Here, we report the structure of the GPRC5D homodimer in complex with a scFv derived from a drug candidate antibody. This structure challenges the earlier speculation regarding the topology of GPRC5D, revealing a greater similarity to the Class C GPCRs albeit with a small extracellular region. Secondly, contrary to classical Class C receptors, GPRC5D, despite lacking an N-terminal VFT domain, is capable of forming stable dimers through the transmembrane helical region. Furthermore, its dimer interface is distinctive from other Class C receptors but retains certain conservation within the GPRC5 subfamily. We propose a "dimer motif" potentially present in the GPRC5 subfamily which all lack the extracellular VFT domain. In addition, through comparison with the structures of mGluR$_2$ in active[32] and inactive states[31], we determined that the structure of GPRC5D, in the absence of any ligand, exhibits an inactive state. Lastly, we revealed the specific binding interface between GPRC5D and scFv, providing a precise template for the development of tool antibodies to guide future antibody drug design.

However, our study still holds several limitations. Firstly, we couldn't ascertain the specific type of downstream G-protein binding, although some researchers have reported a weak Gi signal using BRET2 experiments[25]. Secondly, its activation and signal transduction mechanisms remain elusive. In addition, the widely observed three-layered ionic locks[24,31,37] in Class C GPCRs and motifs associated with receptor activation have shown a lack of conservation in GPRC5D. This implies the existence of a unique activation mechanism for GPRC5D yet to be uncovered. Despite the low homology, we attempted to explore potential activation mechanisms for GPRC5D. Residues 6.50, 6.53, and 6.57 undergo significant conformational changes during the activation of Class C receptors such as mGluR$_2$ and CaSR (the two receptors share relatively high similarity to GPRC5D) (Supplementary Table 2). Activation-related signature movements include the flipping of residue 6.50 in mGluR$_2$[32] (Fig. 3e, f and Supplementary Fig. 6c) and the flipping of residues 6.53 (Fig. 3g) and 6.57 (Supplementary Fig. 6f) in CaSR[33]. We compared these amino acids with GPRC5D and found that although GPRC5D shares similar conformations with inactive states of mGluR$_2$ or CaSR at some residues, they do not completely align, and the sequences of these key residues are not fully conserved in GPRC5D. Finally, despite prior investigations[13] in this direction, GPRC5D still stands as an orphan receptor. Our study did not yield clear indications regarding the identity of endogenous ligands, as no additional density was observed in the orthosteric pocket in the structure, which may be attributed to its inactive state, though.

In summary, our findings unveil a distinctive Class C GPCR, characterized not only by a small N-terminal extracellular region compared to classical Class C receptors but also by a unique dimer interface. The limited sequence similarity to other GPCRs has posed challenges for discovering endogenous ligands and exploring activation mechanisms for GPRC5D. However, the specific binding interface and molecular interaction details between GPRC5D and scFv may potentially expedite the development of targeted antibody therapies for GPRC5D, leveraging computation and AI-aided approaches, to bring hope for the treatment of multiple myeloma.

## Methods

### Protein expression of human GPRC5D and scFv150_18 complex

The gene of human GPRC5D-WT (UniProt ID: Q9NZD1) was synthesized by GenScript and subcloned into the expression vector pFastbac1. The construct contained residues 1–298 of GPRC5D followed by an HRV 3C protease recognition site and a 10x His tag at the C terminus. Haemagglutinin signal peptide, Flag tag, and thermostabilized Escherichia coli apocytochrome b562RIL (BRIL)[41,42] were added on the N-terminus to increase the expression yield of GPRC5D.

The antibody genes used for structural analysis and screening were also synthesized by GenScript. The scFv150_18 genes were cloned into the expression vector pFastbac1, and the Fab genes were cloned into the expression vector pFastbac-dual. The N-terminus of the antibodies was inserted with the GP64 signal peptide.

GPRC5D and scFv were co-expressed in Trichuplusia ni Hi5 insect cells (Invitrogen, B85502), using the Bac-to-Bac baculovirus expression system (Thermo Fisher). Cell cultures were grown in ESF 921 serum-free medium (Expression Systems) to a density of $4 \times 10^6$ cells/mL. For the expression of the GPRC5D and scFv150_18 complex, Trichuplusia ni Hi5 cells were infected at a cell density of $2$–$2.5 \times 10^6$ cells/mL with two separate virus preparations for GPRC5D and scFv150_18 at a ratio of 1:2. After infected by 48 h, the cells were harvested by centrifugation at $1300 \times g$ (Thermo Fisher, H12000) for 20 min and kept frozen at $-80\,°C$ for further usage.

### Purification and formation of GPRC5D-scFv150_18 complex for cryo-EM study

The cell pellets of GPRC5D and scFv complex were thawed and washed with a low-salt buffer containing 10 mM HEPES pH 7.5, 20 mM KCl,

10 mM MgCl$_2$, protease inhibitor cocktail (Roche), and discard the supernatant by centrifugation at 38,000 × $g$ for 30 min. Before solubilization, purified cell pellets were resuspended and incubated with 2 mg/ml iodoacetamide (Sigma) at 4 °C for 30 min. The complex was extracted from the membrane by adding HEPES, NaCl, lauryl maltose neopentyl glycol (LMNG) (Anatrace), and cholesteryl hemisuccinate (CHS, Sigma) to the membrane solution to a final concentration of 50 mM, 500 mM, 1.0% (w/v) and 0.2% (w/v), respectively, and stirred for 3 h at 4 °C. The supernatant was collected by centrifugation at 38,000 × $g$ for 30 min and incubated with TALON IMAC resin (Clontech) and 20 mM imidazole at 4 °C overnight. Then the resin was centrifuged at 800 × $g$ for 10 min and washed with 15 column volumes of buffer I containing 50 mM HEPES pH 7.5, 500 mM NaCl, 5% (v/v) glycerol, 0.05% (w/v) LMNG, 0.01% (w/v) CHS, 10 mM MgCl2, 30 mM imidazole and followed by 15 column volumes of wash buffer II containing 25 mM HEPES pH 7.5, 100 mM NaCl, 5% (v/v) glycerol, 0.03% (w/v) LMNG, 0.006% (w/v) CHS, and 50 mM imidazole. Finally, the protein was eluted using 3 column volumes of an elution buffer containing 25 mM HEPES pH 7.5, 100 mM NaCl, 5% (v/v) glycerol, 0.01% (w/v) LMNG, 0.0002% (w/v) CHS, and 220 mM imidazole. Complexes were loaded onto a Superdex 200 Increase 10/300 column (GE Healthcare) with buffer containing 20 mM HEPES pH 7.4, 100 mM NaCl, 0.00075% (w/v) LMNG, 0.0002% (w/v) CHS to separate the complex from contaminants. Eluted fractions consisting of the GPRC5D-scFv150_18 complex were pooled and concentrated for electron microscopy experiments.

### Expression and purification of IgG for FACS
The IgG gene was synthesized using GenScript and subcloned into the expression vector pcDNA3.4. The IgG for both heavy and light chains were transfected into HEK-293F mammalian cells at a ratio of 1:1. After 72 hours of expression, the supernatant was collected. The supernatant was then passed through a protein A column pre-treated with 20 mM sodium phosphate buffer at pH 7.0. IgG was eluted using 100 mM glycine at pH 3.0. Following neutralization with Tris buffer, the IgG was further purified using a Superdex 200 column. The obtained protein was stored at − 80 °C for future use.

### Preparation of vitrified samples for Cryo-EM study
3.2 μL of the purified GPRC5D-scFv150_18 complex at a concentration of around 3 mg ml$^{-1}$ were applied to glow-discharged 300-mesh Au grids (Quantifoil, R1.2/1.3). The excess sample was removed by blotting with filter paper for 3.5 s before plunge-freezing in liquid ethane using a FEI Vitrobot Mark IV at 100% humidity and 4 °C.

### Cryo-EM data collection
All datasets were collected on a Titan Krios 300 kV electron microscope (Thermo Fisher Scientific, USA) equipped with a GIF Quantum energy filter (20 eV energy slit width, Gatan Inc., USA). The dataset was recorded by a K3 camera (Gatan) at a nominal magnification of 105,000 (calibrated pixel size: 0.832 Å/pixel) and 15 e$^-$/pixel$^2$/s. The movies were recorded using the super-resolution counting mode by SerialEM which applied the beam image shift acquisition method with one image near the edge of each hole. A 50 μm C2 aperture was always inserted during the data collection period. The defocus ranged from − 1.0 to − 2.2 μm. For each movie stack, a total of 40 frames were recorded, yielding a total dose of 60 e$^-$/Å$^2$.

### Cryo-EM image processing
For GPRC5D-scFv150_18 complex, 10,538 movies were recorded and processed with cryoSPARC v.4.2[43]. Patch motion correction was used for beam-induced motion correction. Then, contrast transfer function (CTF) parameters for each dose-weighted micrograph were estimated by patch CTF estimation. Only images with the highest resolution of less than 4 Å were selected for further processing. A total of 9898

images were selected for auto-blob picking, and particles were extracted to do 2D classification. 2D class averages with diverse orientations and clear secondary features were selected as 2D templates for another round of the autopicking process by cryoSPARC. A total of 577,254 particles were selected from 2D classification to generate the initial models. These particles and initial models were used to do 3D classification in heterogeneous refinement in cryoSPARC. 78,948 particles were selected for the final homogeneous refinement followed by nonuniform refinement and local refinement with C2 symmetry in cryoSPARC, resulting in a density map with a nominal resolution of 3.34 Å for the GPRC5D-scFv150_18 complex (determined by gold-standard Fourier shell correlation (FSC), 0.143 criterion). Estimation of the local resolution was performed in cryoSPARC.

### Cryo-EM model building and refinement
The homology models of GPRC5D and scFv150_18 were initially generated by Alphafold[44]. Each part of the target models was docked into the electron microscopy density map using UCSF Chimera[45]. Then, these models were used for model building and refinement against the electron density map. Subsequently, the generated model was manually adjusted in Coot[46] followed by automatic real space refinement in real space in Phenix[47] for several iterations. The model statistics were validated using Phenix[47]. The final refinement statistics are provided in Supplementary Table S1. The cartoons of all structures were generated by PyMOL.

### FACS assay
To measure the binding between GPRC5D and antibody150_18, we performed FACS assay. In brief, HEK293T cells were plated in a 6-well plate. After 2 h, cells were transiently transfected with plasmids encoding WT or mutated GPRC5D using Lipofectamine 2000 reagent (Life Technologies). 48 hours after transfection, 5 μL cells were distributed into a 96-well plate and incubated with 5 μL IgG150_18 of different concentrations for 60 min. Then the cells were incubated with 10 μL goat anti-Human IgG (Fc specific)-FITC (fluorescein isothiocyanate) antibody (Sigma) for 60 min at 4 °C, and then a 9-fold excess of PBS was added to cells. Then we used a Guava flow cytometer to measure the binding between IgG150_18 and GPRC5D or mutants by detecting the fluorescent intensity of FITC using a Guava EasyCyte HT system (Millipore). The Guava Express Plus GRN histogram statistics provide the count, cells/mL, mean fluorescence intensity, and %CV for each population within a marker. Additionally, the % of the total (% of total = (number of cells expressing GPRC5D that are bound by antibodies/number of total cells) × 100) shows the percentage of the data displayed in that plot[41]. Finally, the data of % of the total was used for data analysis. Data were normalized to the expression level. Data were analyzed by nonlinear regression using GraphPad Prism 8.0.

### Reporting summary
Further information on research design is available in the Nature Portfolio Reporting Summary linked to this article.

## Data availability
All relevant data are available from the authors and/or included in the manuscript or Supplementary Information. The atomic coordinates and the electron microscopy maps of the scFv150-18-GPRC5D complex have been deposited in the Protein Data Bank (PDB) under accession code 8YZK, and Electron Microscopy Data Bank (EMDB) under accession code EMD-39696. Source data are provided with this paper.

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

## Acknowledgements

This work was supported by the National Natural Science Foundation of China (32071194 to F.X.), Shanghai Ninth People's Hospital, Shanghai JiaoTong University School of Medicine-Shanghai University of Science and Technology Cross-funded Collaborative Program (JYJC202233 to F.X.), and the Chenguang Program of Shanghai Education Development Foundation and Shanghai Municipal Education Commission (23CGA79 to X.L.). The cryo-EM data were collected at the Bio-Electron Microscopy Facility, ShanghaiTech University, with the assistance of Q.-Q. Sun, L. Wang, and other staff members. We also thank the staff members of the

Cell Expression, Assay, Cloning, and Purification Core Facilities of the iHuman Institute for their support.

## Author contributions

P.Y. performed cloning, protein purification, cryo-EM sample preparation, and data collection. X.L. performed FACS assay, cryo-EM data processing, and model building. L.X. assisted with the IgG purification. L.W. assisted with the cryo-EM data processing. F.L. performed BRET2 assay. J.L. performed protein expression. F.X. designed, coordinated, and supervised the experiments. P.Y., X.L. and F.X. wrote the manuscript. All authors contributed to the data interpretation and preparation of the manuscript.

## Competing interests

The authors declare no competing interests.
