## [Peer Review File · Nature Communications]

The binding mechanism of an anti-multiple myeloma antibody to the human GPRC5D homodimerREVIEWER COMMENTS

Reviewer #1 (Remarks to the Author):

GPRC5D is an emerging target for multiple myeloma that has attracted numerous attentions from the biopharma industry for the development of antibody-based therapeutics. As an atypical Class C GPCR, its structure and oligomerization state remain elusive. In this manuscript, the authors present the first structure of the GPRC5D in complex with a drug-candidate antibody fragment. They elucidate the detailed molecular mechanism underlying the antibody's recognition of GPRC5D, a pivotal insight guiding future antibody or peptide drug design. Through a comprehensive comparison of GPRC5D with representative GPCRs from other families, the study demonstrates that, despite lacking the characteristic VFT domain, the structure of GPRC5D more closely resembles Class C GPCRs than Class A, challenging previous predictions in the absence of structures. Additionally, by structurally comparing GPRC5D with the classical Class C receptor mGlu2, the authors reveal the inactive state of the reported GPRC5D structure. The absence of a conserved ionic lock further clarifies the uniqueness of GPRC5D in the Class C. In addition, despite lacking a significantly large extracellular VFT domain, GPRC5D can still form a dimer, and the authors propose a motif V/I x x E for the dimer interface within the GPRC5 subfamily. This work resolves debates on the structural profile of GPRC5D and provides a detailed antibody binding interface, offering valuable insights for future drug development targeting this bone marrow target. I think it would be worthwhile for publication in Nature Communications. However, there are some specific issues that should be revised in their manuscript before publication.

Some specific issues:

- 1) Line 201-212: The inward movement of TM 6 in this inactive state, is it exclusive to the mGlu2 receptor, or is it observed in other classical Class C GPCRs? A more comprehensive comparison of TM6 in various active and inactive states would enhance the clarity of their results.
- 2) The methods do not involve the purification of IgG, it should be provided.
- 3) The labeling in Fig. 1a is not sufficiently clear. It is suggested to supplement additional annotations related to the extracellular region of the N-terminus in classic C-family receptors, such as the VFT domain.
- 4) It is suggested to add a discussion of potential activation mechanisms of GPRC5D, although the downstream signaling partner is still unclear for this target.

Reviewer #2 (Remarks to the Author):

Yan et al report the cryoEM structure of an orphan class C GPCR, GPRC5D, complexed with a therapeutic antibody in scFv format. The structural analyses together with mutagenesis characterizations not only reveal a distinct dimer arrangement of this atypical class C GPCR, but also illustrate how it's targeted by antibodies of clinical interest. These results would help advance our understanding of class C GPCRs and also help guide rational optimizations for better therapeutic binders.

Here are my suggestions for improving the manuscript:

1. Although the cryoEM map at 3.34Å nominal resolution is of sufficient quality at the 7TM core and dimer interface, the map quality is quite poor at the peripheral TMs, and especially at the antibody-7TM interface and the light-chain half of the scFv. Consequently, the model-building in these problematic regions is not supported by the cryoEM densities. The authors need to improve the cryoEM map by either collecting more data or optimizing data processing and residues should be stubbed to Ala in regions where there is no sufficient density to support side-chain model building.
2. In the analysis of scFv-7TM interface, the authors only mutated residues from N-term, ECL1-3 en bloc without teasing out key interaction(s) that contribute most significantly to antibody binding. I would suggest the authors clearly state what the relevant mutations are either in the text or in figure legend and conduct more detailed point mutations to identify key interactions and find out the extent of how each residue contribute to binding. Furthermore, complementary mutations on the scFv is also needed to validate the binding interface.
3. GPRC5D has been reported to have measurable basal Gi-coupling activity. It would be helpful to find out how the antibody-binding and disruption of 7TM dimer interface as reported by the cryoEM structure affect the basal activity of the receptor. Furthermore, there are interesting interactions at the intracellular half of the 7TM that would seem to limit the canonical shift of TM6 and ICL3 necessary for class C GPCR activation. It would be interesting to find out by mutagenesis how these residues affect basal activity.
4. There are several cholesterol-like densities at the 7TM dimer interface. Does cholesterol binding contribute to dimer formation or affect receptor basal activity? Given cholesterol is known to affect GPCR function, it would be helpful to conduct mutagenesis studies at the binding sites to tease out the functional consequences.

Reviewer #3 (Remarks to the Author):

Yan et al., performed a structural characterization of class C GPCR GPRC5D. GPRC5D is preferentially expressed in differentiating cells that produce hard keratin proteins but also at the surface of multiple myeloma cells, making it as an attractive target for therapies against multiple myeloma. Despite displaying a small N-terminal extracellular domain, the structure of the receptor in complex with pre-clinical single-chain antibody scFv candidate delivers important information. They identify a head-to-head homodimer, an all-marked of class C receptors. Dimerisation is mediated through the 7TMs domain only, whereas the N-terminal domains are not involved in the receptor dimerisation. The authors identify the scFv binding site on the receptor and confirm it by using site-directed mutagenesis. This study is well-conducted, the manuscript is well presented but could be even more straight to the point in some sections.

Line 103-109 the authors state « While in the patent, the antibody binding site is mapped to the N-terminal extracellular region and two extracellular loops (ECL2 and ECL3) (Supplementary Fig. 1c), our structure unveils that the antibody extensively binds to all extracellular regions in GPRC5D, including the N-terminal region, ECL1, ECL2 and ECL3, with an overall interface area of 911 Å² within one copy of the GPRC5D- scFv complex or 1822 Å² for the GPRC5D homodimer ». So, the only difference is the interaction with ECL1 compared to the patent description? I would suggest to rephrase the sentence.

Line 210-212. “Similarly, in GPRC5D, its W6.50 residue resembles that in the inactive state of mGluR2 (Fig. 3g and Supplementary Fig. 6c). Consequently, we propose that GPRC5D adopts an inactive-like conformation.” Having just one residue side-chain rotamer does not sounds enough to me to reach the conclusion that the receptor conformation represents an inactive state. Residues 6.53 might also support this statement, has seen in calcium receptor structure where it faces the inside of the helical bundle. However, conformational changes in class C GPCRs are rather subtles when it comes to compare active and inactive state at the 7TM level.

Line. 234-236. “While the dimer interfaces of most Class C receptors are located in the N-terminal extracellular VFT domain, GPRC5D uniquely relies on the transmembrane region for its dimer interface.” When comparing the surface area with other class C receptor, can you please present which conformation is analyzed for other class C receptors. Indeed, for example 6N51 (in Sup 3 table) represent an active state of mGlu5, and it does not really make sense to compare this interface with the one presented here; and described as an inactive state. Otherwise, it should be described in the table legend. Also, please provide the software used to calculate the surfaces.

Line 237-239. “Examination of the dimer interface reveals that unlike the obligate dimer interface established by disulfide bonds in other Class C GPCRs, there is no covalent bond formed in GPRC5D dimer.” This sentence is not clear.

Line 254-to the end of this paragraph. Since Class C GPCRs commonly function via the formation of homomeric or heteromeric dimers, we further compared the dimer interfaces within the 7TM region across all GPCRs, including reported dimers in Class A, Class C, and Class D receptors³⁶ (Supplementary Fig. 7a). This part and the reference to supplementary figure 7 is difficult to follow;

panel a and b cannot be read properly. This could be shortened since the conclusion is to propose a unique interface specific of GPRC5 subfamily.

Overall, findings bring new information to the field.

We express our gratitude to the three reviewers for their diligent evaluation of our manuscript. Their constructive suggestions and comments have helped us a lot to improve the manuscript during the revision stage. Our point-by-point responses to each reviewer's comments are listed below in blue text. The textural changes in the revised manuscript are shown in red in the marked-up version. All the line numbers indicated in this rebuttal are referred to the ones in the marked-up version.

Reviewer #1:

GPRC5D is an emerging target for multiple myeloma that has attracted numerous attentions from the biopharma industry for the development of antibody-based therapeutics. As an atypical Class C GPCR, its structure and oligomerization state remain elusive. In this manuscript, the authors present the first structure of the GPRC5D in complex with a drug-candidate antibody fragment. They elucidate the detailed molecular mechanism underlying the antibody's recognition of GPRC5D, a pivotal insight guiding future antibody or peptide drug design. Through a comprehensive comparison of GPRC5D with representative GPCRs from other families, the study demonstrates that, despite lacking the characteristic VFT domain, the structure of GPRC5D more closely resembles Class C GPCRs than Class A, challenging previous predictions in the absence of structures. Additionally, by structurally comparing GPRC5D with the classical Class C receptor mGlu2, the authors reveal the inactive state of the reported GPRC5D structure. The absence of a conserved ionic lock further clarifies the uniqueness of GPRC5D in the Class C. In addition, despite lacking a significantly large extracellular VFT domain, GPRC5D can still form a dimer, and the authors propose a motif V/I x x E for the dimer interface within the GPRC5 subfamily. This work resolves debates on the structural profile of GPRC5D and provides a detailed antibody binding interface, offering valuable insights for future drug development targeting this bone marrow target. I think it would be worthwhile for publication in Nature Communications. However, there are some specific issues that should be revised in their manuscript before publication.

Response: We sincerely appreciate the reviewer for providing a positive evaluation and offering constructive comments. In response, we have diligently revised the manuscript and expanded the discussion on the potential activation mechanisms of GPRC5D. Additionally, we performed more comprehensive structural analysis, the details of which are provided below.

Following are few of my specific comments

1. Line 201-212: The inward movement of TM 6 in this inactive state, is it exclusive to the mGlu2 receptor, or is it observed in other classical Class C GPCRs? A more comprehensive comparison of TM6 in various active and inactive states would enhance the clarity of their results.

Response: We appreciate the reviewer's comment. We initially compared the conformational feature of TM6 in GPRC5D with the active-state mGluR₂, as mGluR₂ has the highest sequence similarity to GPRC5D among class C GPCRs. Following the

reviewer’s suggestion, we conducted a more comprehensive comparison of TM6 in different inactive-to-active pairs for class C GPCRs. We compared the TM6 in receptors such as CaSR, mGluR₁, and mGluR₅ (as shown in **Fig. R1a**). Intriguingly, we find a similar **downward movement** of TM6 in mGluR₅ from inactive to active states, akin to the observation in mGluR₂ (as shown in **Fig. R1b**, also in updated **Supplementary Fig. 6d, e**), supporting our conclusion that the GPRC5D structure in our study adopts an inactive-like conformation. In accordance, we revised the wordings in the manuscript, **lines 215-218**, by specifying the feature of TM6 movement as “**downward movement**” from inactive to active states.

Fig. R1 (also shown in revised Supplementary Fig. 6d, e) | Comparison of TM6 in GPRC5D and other Class C GPCRs. a The comparison of the inactive-to-active pair structures of TM6 in CaSR, mGluR₁, mGluR₂, and mGluR₅. **b** The comparison of TM6 between inactive and active states of mGluR₅ and GPRC5D.

2. The methods do not involve the purification of IgG, it should be provided.

Response: We have added purification methods for IgG in the Methods section under the subtitle “**Expression and purification of IgG for FACS**” (**lines 523-530** in the revised manuscript).

3. The labeling in Fig. 1a is not sufficiently clear. It is suggested to supplement additional annotations related to the extracellular region of the N-terminus in classic C-family receptors, such as the VFT domain.

Response: We have added the suggested annotations including “VFT”, “orthosteric binding site” and “allosteric binding site” to each domain in the revised **Fig. 1a**.

4. It is suggested to add a discussion of potential activation mechanisms of GPRC5D, although the downstream signaling partner is still unclear for this target.

Response: We thank the reviewer for the constructive comment. As the downstream signaling partner for GPRC5D remains unclear, we focused on comparing activation-related motifs/residues in GPRC5D with those in other class C GPCRs. Previous studies suggested that residues 6.50, 6.53, and 6.57 undergo significant conformational changes during the activation of receptors such as mGluR₂ and CaSR (the two receptors share relatively high structural similarity to GPRC5D among class C GPCRs; Reviewer #3 also suggests adding a comparison to CaSR), including flipping of residue 6.50 in

mGluR₂ (Fig. 3e, f) and flipping of residues 6.53 and 6.57 in CaSR (Fig. 3g and Supplementary Fig. 6f). However, since these residues lack conservation in GPRC5D, this comparison may not provide a solid understanding of GPRC5D's activation mechanism. In response, we have included a discussion on potential activation mechanisms in the manuscript's discussion section. The new comparison to CaSR is shown in Fig. R2, also in updated Fig. 3g and Supplementary Fig. 6f. The related text can be found in lines 329-339 of the revised manuscript.

Fig R2 (also shown in updated Fig. 3g and Supplementary Fig. 6f) | Comparison of activation-related motifs/residues between GPRC5D and other Class C GPCRs. The comparison of residues 6.53 (a) and 6.57 (b) between inactive CaSR (orange, PDB: 7SIN), active CaSR (cyan, PDB: 7SIM) and GPRC5D (PDB: yellow).

Reviewer #2:

Yan et al report the cryo-EM structure of an orphan class C GPCR, GPRC5D, complexed with a therapeutic antibody in scFv format. The structural analyses together with mutagenesis characterizations not only reveal a distinct dimer arrangement of this atypical class C GPCR, but also illustrate how it's targeted by antibodies of clinical interest. These results would help advance our understanding of class C GPCRs and also help guide rational optimizations for better therapeutic binders.

Response: We appreciate the reviewer's positive feedback and constructive suggestions. We have conducted mutagenesis and functional experiments to address the comments regarding the antibody binding, dimer interface, and potential cholesterol binding site. Additionally, we diligently tried different masking strategies to further refine the cryo-EM map. All these efforts have enhanced the comprehensiveness of the manuscript.

Following are few of my specific comments

1. Although the cryo-EM map at 3.34 Å nominal resolution is of sufficient quality at the 7TM core and dimer interface, the map quality is quite poor at the peripheral TMs, and especially at the antibody-7TM interface and the light-chain half of the scFv. Consequently, the model-building in these problematic regions is not supported by the cryo-EM densities. The authors need to improve the cryo-EM map by either collecting more data or optimizing data processing and residues should be stubbed to Ala in regions where there is no sufficient density to support side-chain model building.

Response: We thank the reviewer for the constructive comment. Over the past several weeks, we have made further efforts to optimize the cryo-EM data processing. We explored two mask methods (one is to focus on the scFv-GPRC5D interface, and the other is to focus on one GPRC5D-scFv monomer) (**Fig. R3b**). Although these methods notably enhanced the density of the light chain portion, the overall density remained largely unchanged (**Fig. R3c**). Despite our best efforts to integrate the post-masked map with the initial map to enhance overall density, technical challenges prevented us from achieving this goal. Consequently, we have included the post-masked map as supplementary material, as merging it with the initial density proved unfeasible.

It is important to emphasize that while the new method and resulting map enhanced certain densities in the antibody regions, they did not alter any of the conclusions drawn in our manuscript. Following the reviewer's suggestion, we have trimmed the side chains for residues where density is not good enough. The final revision to the map and model can be found in the **Supplementary Table 1 and Fig. 1c**. While the overall resolution is adequate (**Fig. R3a**), the side-chain information remains unclear for some areas, which may be attributed to the inherent flexibility of the interface region. Notably, the resolution of the scFv bound to the N-terminus of the receptor is significantly higher than that bound to other extracellular loops. Combining our findings from additional mutagenesis and flow cytometry experiments presented in the revised manuscript (**Fig. 2c, d**), we concluded that the N-terminus of GPRC5D is crucial for antibody binding. This may explain why the density of the interface between the antibody and GPRC5D

was insufficient in other regions.

Fig R3 | The resolution information for the initial map and after applying two mask methods. **a** The initial map used for all the structural analyses in the manuscript. **b** The two maps obtained using different masking methods, with the left one focusing on the scFv and right on one monomer, respectively. **c** Comparison of the antibody regions in the maps obtained from the three methods, with substantial density enhancements at the antibody portion highlighted in red and blue boxes for each monomer.

2. In the analysis of scFv-7TM interface, the authors only mutated residues from N-term, ECL1-3 en bloc without teasing out key interaction(s) that contribute most significantly to antibody binding. I would suggest the authors clearly state what the relevant mutations are either in the text or in figure legend and conduct more detailed point mutations to identify key interactions and find out the extent of how each residue contribute to binding. Furthermore, complementary mutations on the scFv is also needed to validate the binding interface.

Response: We thank the reviewer for the constructive comment. To further define the antibody recognition code on the GPRC5D, we undertook several additional mutations on GPRC5D, including single mutations of S8A, T9A, D11A, L15A, C16A, E83A, P235A, Q236A and combined mutations. We found that C16^{N-Term} in GPRC5D was the most important residue for antibody recognition, since the GPRC5D C16A mutant could no longer be bound by IgG150-18 according to the FACS result (**Fig. R4, also shown in updated Fig. 2d**). At the same time, we cloned and purified several mutant antibodies (**Fig. R4a**) to cross-verify the binding interface. Among which, IgG150-18 with Y228^{HCDR2}A mutation show decreased binding to wild-type GPRC5D (**Fig. R4b**). Therefore, the C16^{N-Term} in GPRC5D and Y228^{HCDR2} in IgG150-18 (C16^{N-Term} is in close contact with Y228^{HCDR2}) are essential for the binding interface. Additionally, mutations affecting residues involved in hydrogen bonding, such as Q236^{ECL3} and T227^{HCDR2}, substantially impacted antibody binding, underscoring the importance of these hydrogen bonds. The relevant analysis has been added to **lines 124-133** of the

revised manuscript. Additionally, based on the reviewer's suggestion, we have annotated relevant mutations in the N-terminus, ECL1, ECL2, and ECL3 in the figure legends (Fig. 2d) to enhance the clarity of our figures. Related textual revisions can be found in lines 116-118 in the manuscript.

Fig. R4 (the panel b is also shown in updated Fig. 2d) | FACS binding assays were performed for mutations on antibody or GPRC5D. a IgG mutants purification (left: SEC results; right: SDS-PAGE results). **b** The term "entire interface mutant" denotes combined mutations at all scFv binding sites on GPRC5D, whereas "IgG all" denotes combined mutations including S244A, T225A, Y226A, Y228A, and Q271A on IgG. The data were replicated three times and normalized to WT GPRC5D. Significance was determined by Ordinary one-way analysis of variance (ANOVA), followed by Dunnett's multiple comparisons test (** $P < 0.001$, * $P < 0.05$; n.s., not significant).

3. GPRC5D has been reported to have measurable basal Gi-coupling activity. It would be helpful to find out how the antibody-binding and disruption of 7TM dimer interface as reported by the cryoEM structure affect the basal activity of the receptor. Furthermore, there are interesting interactions at the intracellular half of the 7TM that would seem to limit the canonical shift of TM6 and ICL3 necessary for class C GPCR activation. It would be interesting to find out by mutagenesis how these residues affect basal activity.

Response: We thank the reviewer for the constructive comment. Although there have been reports indicating that GPRC5D may possess basal Gi coupling activity by Watkins et al, *British journal of pharmacology* 2020, the evidence remains inconclusive. We therefore investigated the basal activity of GPRC5D using BRET2 experiments

according to the established protocols, which have been successfully employed to assess basal activity in orphan receptors (such as orphan GPR21, as reported by Lin et al, *Nature Communications* 2020; and orphan GPR20, as reported by Lin et al, *Cell Discovery* 2023). We meticulously optimized various parameters such as plasmid transfection amounts, expression time and cell numbers, and used GPR20—a receptor with high basal activity of Gi coupling—as a control. However, our experiments did not reveal any basal activity in GPRC5D (**Fig. R5a**). Despite the lack of observed basal activity in GPRC5D, we tested the impact of mutations at the antibody interface (**Fig. R5b**) and dimer interface (**Fig. R5c**) on its potential Gi-coupling activity. However, we did not observe any mutations that could enhance its basal activity. Given the absence of observed basal activity in GPRC5D and the lack of mutations that influenced its basal activity, we have chosen not to extensively discuss basal activity in this manuscript.

Fig. R5 | BRET2 experiments were performed to evaluate the basal activity of GPRC5D and the effects of mutations at the dimer and antibody interfaces on its basal activity. a BRET2 experiments were conducted to measure the basal activity of GPRC5D. **b** BRET2 experiments were performed to assess the impact of dimer mutations on basal activity. **c** BRET2 experiments were performed to assess the impact of antibody mutations on basal activity. GPR20 is used as a positive control.

4. There are several cholesterol-like densities at the 7TM dimer interface. Does cholesterol binding contribute to dimer formation or affect receptor basal activity? Given cholesterol is known to affect GPCR function, it would be helpful to conduct mutagenesis studies at the binding sites to tease out the functional consequences.

Response: We thank the reviewer for the constructive comment. We attempted to fit cholesterol molecules into the density located at the dimer interface on the intracellular side. Based on this binding site, we made mutations (V115A, I132A, L183A, V115A+L183A, and the combined mutations (named as All CHS mutation)) and characterized the dimer formation through size-exclusion chromatography (SEC) using purified GPR52 as a monomer control. We found that the dimerization of GPRC5D was not affected (**Fig. R6a**). Additionally, although GPRC5D did not exhibit detectable basal BRET activity, we also tested the effects of mutations at the cholesterol binding site on the receptor's basal activity and observed that none of the mutations could enhance its basal activity (**Fig. R6b**). Since mutations at the cholesterol binding site had no effect on receptor dimerization or function, and the density is not sufficient to fit a cholesterol molecule, we have chosen not to comment on the cholesterol molecules in this manuscript.

Fig R6 | Mutations of interacting residues within the potential cholesterol pocket were assessed for their effects on dimer formation and basal activity. a The comparison of SEC retention time among different Cholesterol pocket mutants. Black curve represents the WT GPRC5D, which exhibits dimer retention time. GPR52 serves as the monomeric control. **b** BRET2 experiments were performed to assess the impact of Cholesterol pocket mutations on basal activity.

Reviewer #3:

Yan et al., performed a structural characterization of class C GPCR GPRC5D. GPRC5D is preferentially expressed in differentiating cells that produce hard keratin proteins but also at the surface of multiple myeloma cells, making it as an attractive target for therapies against multiple myeloma. Despite displaying a small N-terminal extracellular domain, the structure of the receptor in complex with pre-clinical single-chain antibody scFv candidate delivers important information. They identify a head-to-head homodimer, an all-marked of class C receptors. Dimerisation is mediated through the 7TMs domain only, whereas the N-terminal domains are not involved in the receptor dimerisation. The authors identify the scFv binding site on the receptor and confirm it by using site-directed mutagenesis. This study is well-conducted, the manuscript is well presented but could be even more straight to the point in some sections.

Response: We thank the reviewer for the positive evaluation and constructive comments. We appreciate the insightful feedback on the manuscript content and certain figures. In response, we have addressed unclear content and improved the clarity of the figures, which resulted in a more comprehensive manuscript overall.

Following are few of my specific comments.

1. Line 103-109 the authors state “While in the patent, the antibody binding site is mapped to the N-terminal extracellular region and two extracellular loops (ECL2 and ECL3) (Supplementary Fig. 1c), our structure unveils that the antibody extensively binds to all extracellular regions in GPRC5D, including the N-terminal region, ECL1, ECL2 and ECL3, with an overall interface area of 911 Å² within one copy of the GPRC5D- scFv complex or 1822 Å² for the GPRC5D homodimer”. So, the only difference is the interaction with ECL1 compared to the patent description? I would suggest to rephrase the sentence.

Response: We thank the reviewer for the constructive comment. For better clarity, we removed the inconclusive description of the antibody binding site in the patent, and focused solely on the observation from our structure.

2. Line 210-212. “Similarly, in GPRC5D, its W6.50 residue resembles that in the inactive state of mGluR2 (Fig. 3g and Supplementary Fig. 6c). Consequently, we propose that GPRC5D adopts an inactive-like conformation.” Having just one residue side-chain rotamer does not sounds enough to me to reach the conclusion that the receptor conformation represents an inactive state. Residues 6.53 might also support this statement, has seen in calcium receptor structure where it faces the inside of the helical bundle. However, conformational changes in class C GPCRs are rather subtle when it comes to compare active and inactive state at the 7TM level.

Response: We appreciate the constructive feedback from the reviewer. We further compared residue 6.53 in GPRC5D with CaSR and found that the GPRC5D structure resembles to the inactive state of CaSR, with residue 6.53 pointing towards the helical center. We have incorporated this finding into the manuscript (as shown in **Fig. R7a**, also in updated Fig. 3g). Additionally, based on the reviewer’s suggestion, we have performed the comparison of residues on the 7TM region that play crucial roles during

activation or exhibit significant conformational changes (as shown in **Fig. R7b**, also in updated **Supplementary Fig. 6f**). The results support an “inactive-like” conformation of GPRC5D in our structure. This revision can also be found in **lines 224-228** in the revised manuscript.

Fig. R7 (also in updated **Fig. 3g** and **Supplementary Fig. 6f**) | **Comparison of TM6 between GPRC5D and other Class C GPCRs.** **a, b** The comparison of residues 6.53 and 6.57 in inactive CaSR (orange, PDB: 7SIN), active CaSR (cyan, PDB: 7SIM) and GPRC5D (yellow).

3. Line. 234-236. “While the dimer interfaces of most Class C receptors are located in the N-terminal extracellular VFT domain, GPRC5D uniquely relies on the transmembrane region for its dimer interface.” When comparing the surface area with other class C receptor, can you please present which conformation is analyzed for other class C receptors. Indeed, for example 6N51 (in Sup 3 table) represent an active state of mGlu5, and it does not really make sense to compare this interface with the one presented here; and described as an inactive state. Otherwise, it should be described in the table legend. Also, please provide the software used to calculate the surfaces.

Response: We appreciate the comment from the reviewer. We have removed the irrelevant comparisons with the active conformation in **Supplementary Table 3**. Additionally, based on the reviewer’s suggestion, we have conducted more structural analysis on all the structures mentioned in the table and calculated the dimer interface areas (see **lines 278-282**, **Supplementary Fig. 7a**). Furthermore, we have included descriptions of the conformational states in **Supplementary Table 3** and added the

software (PyMOL 2.5.4) used for surface area calculations in the table footnotes.

4. Line 237-239. “Examination of the dimer interface reveals that unlike the obligate dimer interface established by disulfide bonds in other Class C GPCRs, there is no covalent bond formed in GPRC5D dimer.” This sentence is not clear.

Response: We have reorganized the sentence to remove vague expressions, providing a clearer description that “the dimer interface of GPRC5D is mediated by two pairs of hydrogen bonds and a series of hydrophobic interactions” in lines 258-260 in the revised manuscript.

5. Line 254-to the end of this paragraph. “Since Class C GPCRs commonly function via the formation of homomeric or heteromeric dimers, we further compared the dimer interfaces within the 7TM region across all GPCRs, including reported dimers in Class A, Class C, and Class D receptors³⁶ (Supplementary Fig. 7a).” This part and the reference to supplementary figure 7 is difficult to follow; panel a and b cannot be read properly. This could be shortened since the conclusion is to propose a unique interface specific of GPRC5 subfamily.

Response: We appreciate the constructive feedback from the reviewer. Based on the reviewer’s suggestion, we have revised the description of this section (lines 278-282). We removed the comparison of GPRC5D with other families of GPCRs and retained only the comparison with the class C receptors to emphasize the unique properties of the GPRC5D dimer interface. Additionally, in Supplementary Fig. 7a, we added arrows to indicate rotation, reflecting the differences in dimer interface among different receptors. We also clearly labeled each receptor we compared, making the entire figure more comprehensible (as shown in Fig. R8, also updated Supplementary Fig. 7a). Furthermore, we have clarified in the figure caption that, in order to illustrate the sharp differences of the dimer arrangement and interface between GPRC5D and canonical class C GPCRs, we aligned one subunit of the dimers in representative class C GPCRs with an anchoring subunit of GPRC5D to observe the divergent orientation of the other subunit. Moreover, to enhance the clarity of our dimer interface comparison, we added boundaries for all receptors in Fig. 4c, allowing readers to better discern the differences of dimer interface between GPRC5D and other class C receptors.

Fig. R8 (as updated Supplementary Fig. 7a) | The comparison of GPRC5D dimer interface with other inactive-state receptors. a The comparison between the dimer interface of GPRC5D and other Class C GPCRs is illustrated by aligning one subunit of the dimer with GPRC5D to demonstrate differences in the dimer interface. GPRC5D is depicted in yellow, GPR158 in orange (PDB: 7EWL), CaSR in pink (PDB: 7SIN), mGluR₁ in blue (PDB: 4OR2), mGluR₂ in green (PDB: 7MTQ), and GABA_B in cyan (PDB: 7C7S). The left side displays an extracellular view, while the right side represents an intracellular view.

REVIEWERS' COMMENTS

Reviewer #1 (Remarks to the Author):

The concerns the reviewer raised have been satisfactorily addressed, and there are no more questions.

Reviewer #2 (Remarks to the Author):

The authors have sufficiently addressed my comments.

Reviewer #4 (Remarks to the Author):

The authors have addressed all my comments.

I do have however one more point that should be addressed in the manuscript:

Line 213: In mGluR2, the TM6 undergoes a downward shift by half a helix turn from inactive to active states (Fig. 3e and Supplementary Fig. 6b). Previous studies suggest that this TM6 movement is a characteristic of Class C receptor activation³¹. Similarly, this downward movement of TM6 is also observed in the active-state mGluR5 (Supplementary Fig. 6d, e). A closer comparison of GPRC5D with mGluR2 revealed a noteworthy rotational conformational change in the highly conserved residue W6.50 of the Class C GPCR.

Here only one study is cited for justifying of the TM6 movement as a characteristic of Class C receptor activation. Then when considering such movement, Seven et al, 2021 (Nature) reported that TM6 is moving upward in the G protein coupled 7TM, as well as for the CaSR. In mGlu5, it is described that TM6 moves outward upon PAM binding, although there is no structure with bound G protein. Here one has to be careful when aligning the receptor dimer for comparing the 7TM. In my opinion, it might be wise to tone down the movement of H6 as a feature of activation for GPRC5D, especially if there is not much information about the receptor signalling and no activate state structure.

We express our gratitude to the reviewers for their diligent evaluations of our manuscript. Their constructive suggestions and comments have helped us a lot to improve the manuscript during the revision stage. Our response to Reviewer#4's comment is listed below in blue text. The textural changes in the revised manuscript are shown in red in the marked-up version. All the line numbers indicated in this rebuttal are referred to the ones in the marked-up version.

Reviewer #4 (Remarks to the Author):

The authors have addressed all my comments.

I do have however one more point that should be addressed in the manuscript:

Line 213: In mGluR₂, the TM6 undergoes a downward shift by half a helix turn from inactive to active states (Fig. 3e and Supplementary Fig. 6b). Previous studies suggest that this TM6 movement is a characteristic of Class C receptor activation³¹. Similarly, this downward movement of TM6 is also observed in the active-state mGluR5 (Supplementary Fig. 6d, e). A closer comparison of GPRC5D with mGluR₂ revealed a noteworthy rotational conformational change in the highly conserved residue W^{6.50} of the Class C GPCR.

Here only one study is cited for justifying of the TM6 movement as a characteristic of Class C receptor activation. Then when considering such movement, Seven et al, 2021 (Nature) reported that TM6 is moving upward in the G protein coupled 7TM, as well as for the CaSR. In mGlu₅, it is described that TM6 moves outward upon PAM binding, although there is no structure with bound G protein. Here one has to be careful when aligning the receptor dimer for comparing the 7TM. In my opinion, it might be wise to tone down the movement of H6 as a feature of activation for GPRC5D, especially if there is not much information about the receptor signalling and no activate state structure.

Response: We thank the reviewer for the comment. The comparative analysis of the residue 6.53 on TM6 with CaSR was previously suggested by one of the other reviewers. However, we agree that the sentences describing the movement of TM6 in GPRC5D should be toned down. Based on this suggestion, we have made the following minor revisions to the text: 1) we **deleted the sentences from lines 218 to 223**: “In the structure of mGluR₂ bound to Gi (PDB: 7E9G), ... leading to the downward movement of TM6”. 2) we **toned down the sentences from lines 224 to 229**. 3) we added some

discussion on **lines 230-235**: “However, it is noteworthy that Class C receptors exhibit variations in the movements of TM6 during activation. For instance, TM6 exhibits upward movement in mGluR2 and CaSR, and outward rotation in mGluR5. Although we have comparatively analyzed the structural features of GPRC5D, its activation mechanism remains to be further investigated due to the lack of an active state structure and the fact that it is an atypical Class C GPCR”. 4) we cited related references as the reviewer suggested.